# Simulation of the Compaction Behavior and the Water Permeability Evolution of Broken Rock Masses of Different Shapes in a Goaf

**Yuxi Guo, Yan Qin * , Ping Chen and Nengxiong Xu**

School of Engineering and Technology, China University of Geosciences (Beijing), Xueyuan Road 29, Beijing 100083, China
* Correspondence: qinyancugb@cugb.edu.cn

**Abstract:** The rock mass in the caving zone of a goaf is relatively broken and considered a porous medium. Additionally, it has the characteristics of irregular size and shape and sharp edges, and it is easy to break. In the process of caving zone compaction, the shape characteristic of a broken rock mass is one of the most important factors affecting the evolution of the compaction characteristics and the water permeability of the caving zone. Through discrete element numerical simulation and theoretical research, the influence of the shape characteristic on compaction characteristics and the water permeability of a broken rock mass is analyzed. The research results are as follows: (1) The number of edges on a caved broken rock mass is negatively correlated with the strain limit of compaction, the initial void ratio and the final breaking ratio. It is positively correlated with the deformation modulus and the residual dilatancy coefficient. (2) The smaller the amount of edges on the broken rock mass, the more obviously the rotation movement occurs during compaction. (3) The smaller the number of edges on the broken rock mass, the faster the decline in the rate of the water permeability, and the lower the water permeability at the final stable stage. (4) With an increasing number of broken rock mass edges, the total strain energy and the dissipative strain energy of caved broken rock masses show a decreasing trend, while the elastic strain energy shows a growing trend.

**Keywords:** caving rock masses; shape characteristics; compaction characteristics; fracture rate; energy evolution; water permeability

## 1. Introduction

After underground coal mining, the overburden rock masses in a goaf separately shape a caving zone, fissure zone and bent subsidence zone from the bottom to the top. Rocks in the goaf's caving zone are relatively fractured and considered porous media with pores, fissures and void triple pore structures [1,2]. A caving rock mass has the characteristics of irregular size and shape, sharp edges and corners, internal defects and fragility. The mechanical properties and the water permeability of a broken rock are closely related to its shape, which leads to differences in the internal defects and crushing conditions of the broken rock. With the continuous advancement of the working face, the continuous subsidence of the overlying strata will cause the compaction of the goaf's caving zone, which will result in changes in parameters such as the stress, the porosity, the coefficient of crushing expansion, the crushing rate, the energy dissipation of the crushed rock and the water permeability [3,4]. Therefore, the influence of the shape characteristics of the broken rock mass in the caving zone of a goaf on the compaction characteristics and seepage characteristics has been studied. It is of great significance to the safe and efficient production of coal mining, the storage and utilization of resources such as mine water in the goaf, environmental and ecological protection, and the mechanism of overburden movement [5,6].

A large number of laboratory tests have been carried out on the compaction and the water permeability of caving broken rock masses. Figure 1 shows the compaction stress–strain curves [7–13] of broken rocks such as gangue, mudstone and shale with different particle size gradations. The stress–strain curves of caving rock masses with different particle sizes and gradations show approximately exponential relationships during compaction. The strain at the initial stage of compaction rapidly increases with stress, and the growth rate of strain gradually decreases with increasing stress. Fan et al. [14] and Zhang et al. [15] performed laboratory tests to divide the compaction process of caving rock masses into three stages: the initial compaction stage, the gradual compaction stage and the stable compaction stage. The porosity and the water permeability gradually decrease with the compaction stage. The internal relationship between the compaction stress, the real-time compaction and the water permeability at different locations of the caving zone has been quantitatively analyzed. Wang et al. [7] performed compression tests on broken rocks with different particle sizes and showed that the strain growth rate of large particle size specimens is smaller than that of small particle size specimens at the beginning of loading. With increasing stress, the strain growth rate of large particle size specimens exceeds that of small particle size specimens. Su et al. [10] carried out a compaction test on broken mudstone from a roof to verify the influence rule of particle size on strain. It was concluded that the radial strain of the mixed block with 20% uniformity mixing of different block diameters was small and that the compression resistance was improved. Huang et al. [16] analyzed the influence of the gradation and confining pressure conditions of caving rock masses on their deformation and failure characteristics. It was concluded that the maximum bearing stress of caving rock masses is sensitive to the confining pressure condition and insensitive to the particle size and gradation. The maximum bearing stress linearly increases with increasing confining pressure, while the degree of fracture is significantly affected by the gradation of the caving rock mass but almost independent of the confining pressure condition. Liu et al. [17] and Qin et al. [18] carried out the lateral limit compression test on broken rock and found that the sectional modulus, tangential modulus, porosity, expansion coefficient and compactness of caving rock masses are negatively correlated with the Talbol index.

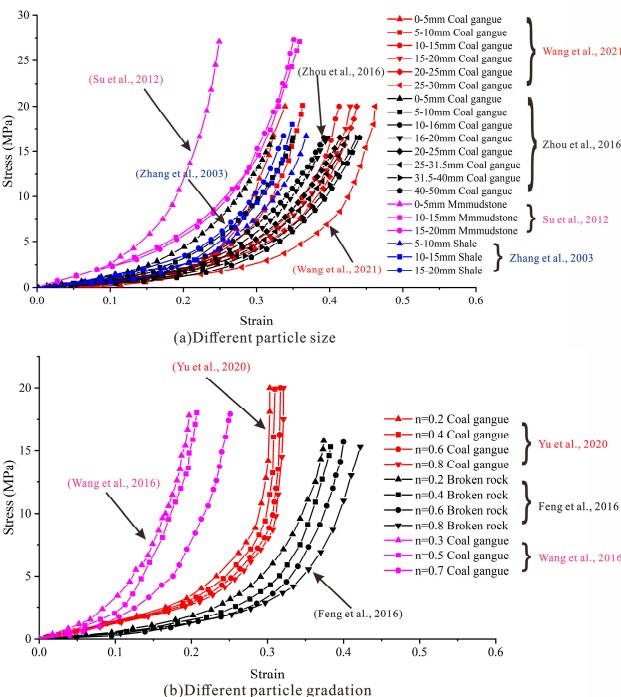

**Figure 1.** Compaction stress and strain relationship of broken rock [7–13].

With regard to the influence of shape characteristics on the compaction characteristics and the water permeability of broken rocks, relevant research uses numerical simulation methods to classify the morphology of caving rock masses into ellipsoids or ellipsoids [19], and polygons or polyhedrons [20] of arbitrary shape, etc. Most scholars use the complex geometrical shape of a single particle to simulate real caving rock masses. Few people bond round particles together in a given form to form a "particle cluster" to represent the irregular shape of the caving rock masses [21]. Liu et al. [22] adopted the agglomeration method to simulate the true triaxial value, which shows agreement with the test results. Lee et al. [23] simulated real particles of earth–rock mixtures with polyhedron units and carried out triaxial compression numerical tests under different confining pressures and initial porosities. The deviation stress curve and volume strain curve were obtained, which indicated the feasibility of this simulation method. Fan and Liu [14] simulated the water permeability process of compacted rock masses based on the cubic mechanism. The simulation showed that, in the initial compaction stage, the particle elastic modulus has no contribution to the overall compaction and permeability reduction. However, for a fully compacted goaf, the particle elastic modulus controls the evolution of water permeability.

To sum up, a substantial amount of research has been performed on different particle sizes, gradations, macro-micromechanical properties and other aspects in the process of rock compaction in goaf caved zones, and prolific research results have been achieved. However, in actual engineering, the compaction characteristics and the water permeability of rock masses are closely related to the complex shapes of rock masses [24,25]. In the existing research results, the indoor test uses the crusher to break the rock, without considering the impact of the shapes. In the numerical simulation, most of the research is assumed to be on a single shape, and there have not been enough articles discussing whether or not shape has effects on compaction characteristics and water permeability. In this paper, the polar coordinate generation method is used to determine the vertex of the polygon and to construct the stochastic morphological characteristics of the caved rock masses in a goaf. By simulating side-limited compression and crushing tests on caved rock masses with different prisms, the relationship between the stress–strain curve and the deformation modulus, the porosity, the water permeability, the crushing coefficient and the crushing rate of caved rock masses with different shapes during the compaction process is studied. The differences in internal particle motion states, water permeability evolution mechanisms, force chain evolution mechanisms and the energy evolution characteristics in the compaction process of broken rock masses with different shapes are obtained.

## 2. Simulation Method

### 2.1. Generation Method of the Rock Mass Model with Different Morphologies

The particle breakage and filling conditions of different shapes of broken rock mass are different in the compaction process, resulting in differences in the compaction characteristics and the water permeability of broken rock masses. Therefore, it is necessary to study the fracture state and fracture development of broken rock masses. The finite element software cannot simulate fracture and fracture development during the fracture process. A discrete element software is usually used for simulation. A rock mechanics simulation software based on discrete element mainly includes UDEC and PFC. The Voronoi polygon model is commonly used in UDEC for simulation. However, the irregular polygon cannot be further broken. It is difficult for it to conform to the rebreaking characteristics of broken rock masses during compaction in the caving zone of a goaf. A PFC simulation of particle breakage usually uses the debris replacement method and the bonding combination method. Because the samples before and after fragmentation by the debris replacement method are round particles, it is difficult to simulate the shape of complex particles, and complex shapes cannot be considered. In order to study the different shapes of broken rock mass, the PFC bond combination method to simulate shapes was used. By setting the broken rock mass template with different shapes in advance, and then giving a parallel bond model to build a cluster of particles to simulate the broken process of the broken rock mass, it

was assumed that a broken rock mass is a particle aggregate. When the contact force of the internal unit of the aggregate reaches its ultimate strength, the contact fracture, parallel adhesion degenerates into linear contact, and then forms a number of broken particles. Based on this, our study adopted a discrete element particle flow numerical simulation to study the compaction of broken rock masses. In this study, the bonding group method was used to simulate the crushing process of crushed stone by constructing a cluster particle cluster. Different shapes of caving rock masses were obtained by changing the number of edges and sphericity of particle clusters. Firstly, the boundary of the model was generated by a wall, and the specific circular gravel template required by oneself was generated within the boundary of the model. At this time, a pebble had only one circular particle. The information, such as the center location and radius of the generated circular gravel template, was recorded and stored, and then these circular gravel templates were deleted and finer grain circular units were generated within the model boundary. These units were used to bond to form granular clusters of rubble. For information comparison, if the small particle unit was inside the previous round gravel template, it was saved, and if the small particle unit was outside the previous round gravel template, it was deleted, and then the target particle cluster gravel could be obtained. The position coordinates were randomly placed within the model range by the random seed dropping method, then the pressure plate was controlled to be settled, and the compaction process of different caving rock masses was simulated. The realization methods of the edges and corners of different caving rocks were as follows:

First, given the radius of the caving rock masses, the circular caving rock masses was generated, and then the vertex of the polygon was determined by the polar coordinate method to ensure that the overall size of the caving rock masses was the same, as shown in Formulas (1) and (2):

$$\theta_k = \frac{2\pi[1 + (2b_k - 1)\delta]}{n} \tag{1}$$

where $\theta_k$ is the angle corresponding to the *K*-edge of the polygon, $b_k$ is a random number in [0, 1], $\delta$ is a constant less than 1, and $n$ is the number of edges of a polygon. After determining the number of polygon edges, to ensure $n$, the sum of $\theta_k$ was set to $2\pi$, which involved a pair of $\theta_k$ to perform the correction.

$$\overline{\theta}_k = \theta_k \left( \frac{2\pi}{\sum_{j=1}^{n} \theta_j} \right) n \tag{2}$$

where $\overline{\theta}_k$ is the angle corresponding to the modified rear edge length. After determining the corresponding angle of each side of the polygon and $2\pi$, the vertex coordinates of the polygon were determined according to the polar coordinates, as shown in Formulas (3) and (4):

$$\begin{cases} x_k = x_0 + r\cos(\alpha + \theta') \\ y_k = y_0 + r\sin(\alpha + \theta') \end{cases} \tag{3}$$

$$\theta' = \sum_{k=1}^{n} \overline{\theta}_k \tag{4}$$

where $\alpha$ is the number of [0, $\pi/2$] used to ensure that the vertex positions are random; $x_0$ and $y_0$ are vertex coordinates of the inner circle of a polygon, representing the horizontal coordinates and vertical coordinates, respectively, of the position of the caving rock mass.

In this study, the method of the randomly generating polygon vertices in the polar coordinates was written as the FISH function. Then, the geometry function was introduced as a template. Geometry has the functions of creating edges, generating geometry, creating polygons, rotating and copying. Thus, the generation of broken rock masses of different shapes could be conducted. Figure 2 shows the process of 2D caving rock mass generation. The steps of pentagonal shape feature generation are presented as an example: (1) five nodes of the polygon were created by commands according to the above formulas; (2) after the random vertex was generated, the points were connected in a line to form the edge

of the polygon in-sequence; and (3) to reduce the error to meet the requirements, the points were connected to form lines, and the triangular mesh was generated. A complete polygonal geometry caving rock mass template was generated.

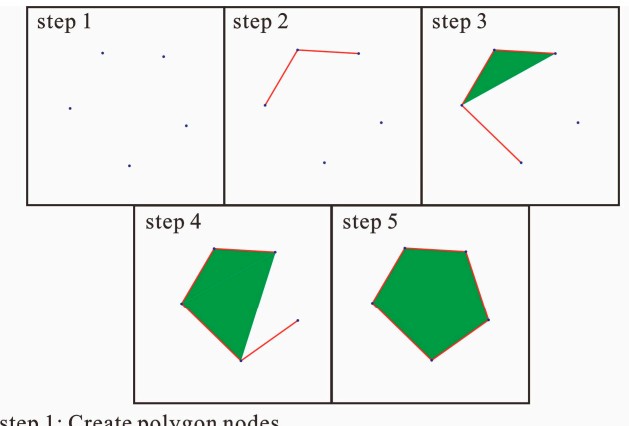

step 1: Create polygon nodes.
step 2~step 5: The edges of a polygon that are formed sequentially.

**Figure 2.** Generation process of 2D caving rock mass.

The particles in the geometry function were defined as the caving rock mass group, the particles between the caving rock masses were defined as other groups, and the other groups were deleted. Based on this method, different shapes of caving rock masses in the two-dimensional case were obtained in order to obtain the change trend of the compaction characteristics and the water permeability of the broken rock masses of different shapes. Therefore, the study selected quadrilateral, pentagonal, hexagonal, octagonal and decagonal broken rock mass shapes for research, as shown in Figure 3.

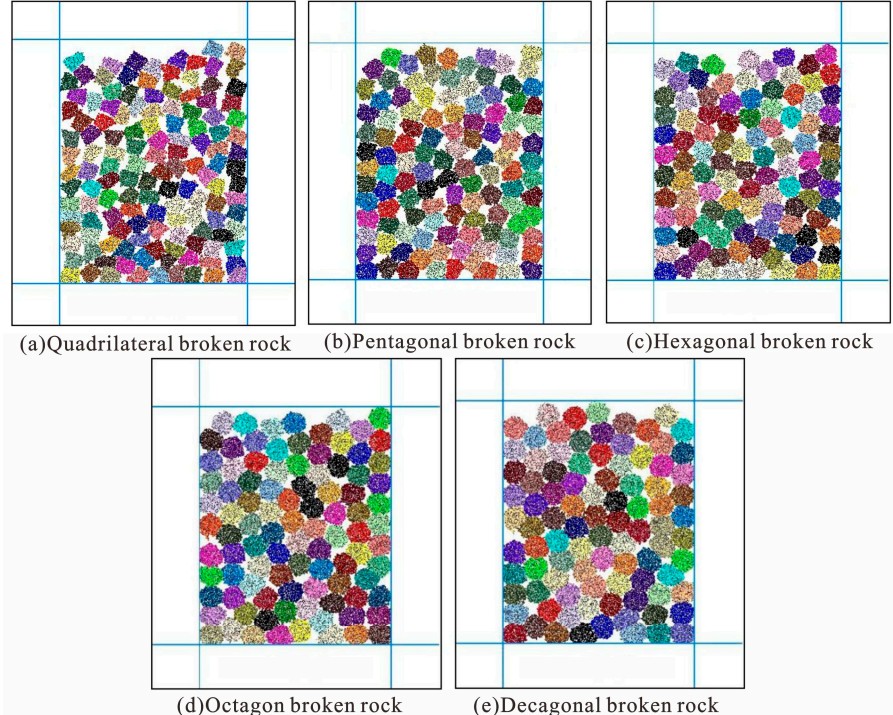

**Figure 3.** Schematic of caving rock mass in different shapes.

### 2.2. Establishment of the Compaction Characteristics and the Water Permeability Test Model of Broken Rock Masses

The use of 3D simulation of broken rock masses with different shapes may lead to dimensional disasters, such as the selection of 3D shapes. Due to the large number of particles in the calculation, the effects of calculation efficiency, the visual observation of particle breakage and possible dimensional disasters in 3D simulation were considered. In this study, 2D particle flow was used for analysis. The particle flow procedure was divided into five steps: model establishment, the generation of caving rock masses, self-weight balance, loading, and monitoring the water permeability and the compaction characteristics. The detailed flow is shown in Figure 4.

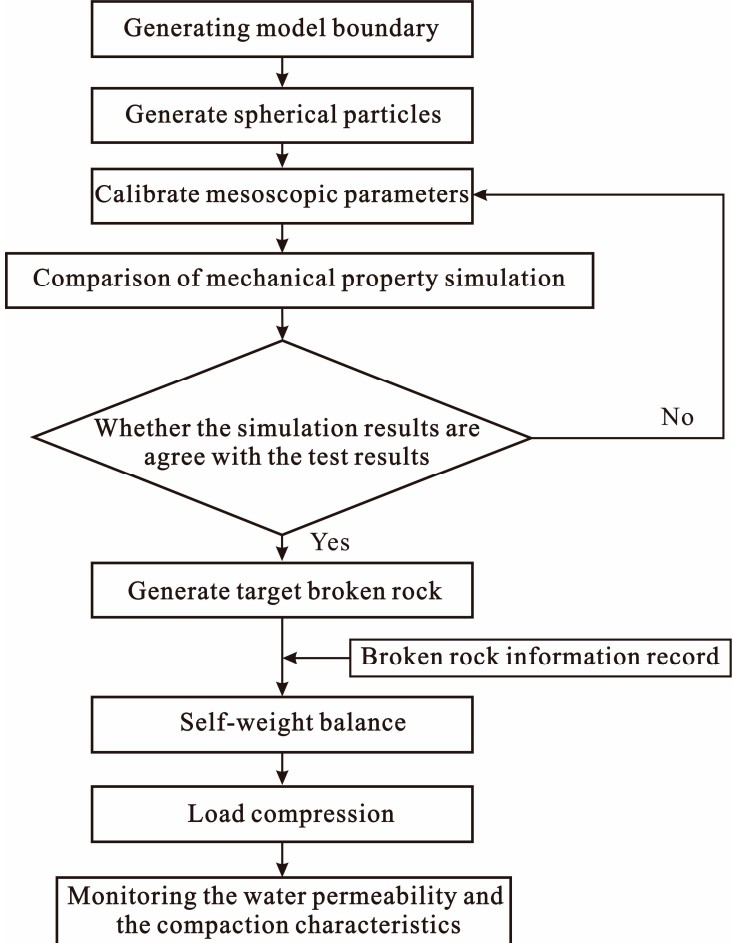

**Figure 4.** Flow chart.

Firstly, a model with a wall as the model boundary was generated. A parallel bond model was assigned between particles. The parameters of Su et al. [10] in the compaction test on crushed rock of a coal seam roof were selected for simulation. The model size restraint wall height was 195 mm, and the width was 160 mm. Displacement control was employed to apply the load, and the radius of the caving rock masses was 10 mm. The number and size of particles have a great influence on mechanical properties and calculation time, according to previous mechanical tests in the laboratory, the Back Propagation neural network parameter inversion method, numerical image processing and other technologies [26], the comprehensive calculation rate, simulation effect and the influence of rock mass morphology. The final particle size was 0.5 mm, with $L/R_{min}$ = 320, $R_{max}/R_{min}$ = 1.66, and $R_{max}$ = 0.83 mm.

Sandy mudstone from the top of a coal seam was utilized in this simulation. The main macro parameters included tensile strength ($\sigma_t$), compressive strength ($\sigma_c$), deformation

modulus ($E_T$), Poisson ratio's ($v$), bond strength ($c$) and friction angle ($\varphi$). The physical and mechanical parameters of this rock are shown in Table 1 [10].

**Table 1.** Physical and mechanical parameters of rock [10].

| Lithology | $\rho$ (kg/m³) | $\sigma_t$ (MPa) | $\sigma_c$ (MPa) | $E_T$ (GPa) | $v$ | $c$ (MPa) | $\varphi$ (°) |
|---|---|---|---|---|---|---|---|
| Sandy mudstone | 2690 | 6.5 | 71.8 | 25.3 | 0.25 | 28.3 | 33.8 |

The complete meso-parameters of the rock samples used for simulation were obtained by parameter inversion, as shown in Table 2.

**Table 2.** Calibrated mesoscopic parameters of caved rock mass.

| Mesoscopic Parameters | Parameter Meaning | Parameter Value |
|---|---|---|
| $\rho_s$ | Density (kg/m³) | 2690 |
| $\mu_s$ | Friction coefficient | 0.50 |
| $L/R_{min}$ | Size ratio | 320.00 |
| $R_{max}/R_{min}$ | Maximum and minimum particle ratio | 1.66 |
| deform_emod | Linear contact modulus (GPa) | 4.37 |
| pb_deform_emod | Effective modulus of parallel bond (GPa) | 25.56 |
| pb_ten | Tangential strength (MPa) | 43.00 |
| pb_coh | Normal strength (MPa) | 52.00 |
| kratio | Stiffness ratio | 1.80 |

To ensure the accuracy of the model parameters determined by the two-dimensional simulation method, according to the compaction test of the caving rock masses of Su et al. [10], a 3D particle flow numerical simulation sample was established for the caving rock masses with a particle size in the range of 10–15 mm. Firstly, the uniaxial tensile test was used to calibrate the effective modulus of the parallel bond. Then, the effective modulus of the parallel bond was fixed and the linear contact modulus, the stiffness ratio, the tangential strength and the normal strength parameters were calibrated by uniaxial compression. The side-limited compaction stress–strain curve of the 3D caving rock mass model was obtained as shown in Figure 5. The stress–strain curve obtained by numerical simulation is in agreement with the test data. The accuracy of the model parameters determined by the two-dimensional simulation method was verified. Due to the large number of particles in the calculation, two-dimensional particle flow was adopted for analysis considering the effects of calculation efficiency, the visual observation of particle breakage and possible dimensional disasters in 3D simulation.

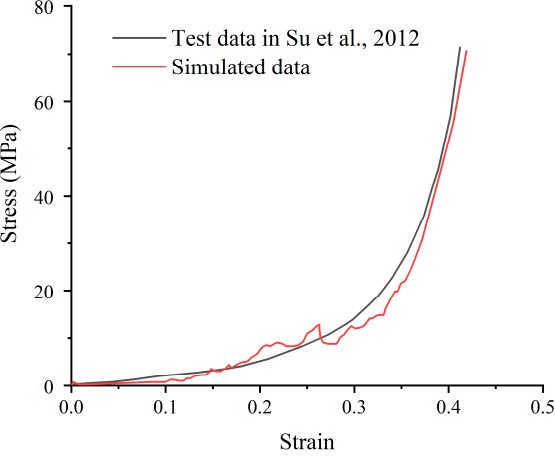

**Figure 5.** Comparison of simulation results [10].

The second step was to generate the caving rock masses. According to the method of rock mass generation mentioned above, the vertex of the polygon was determined by the polar coordinate generation method, and different rock mass models with different shapes were obtained, as shown in Figure 6.

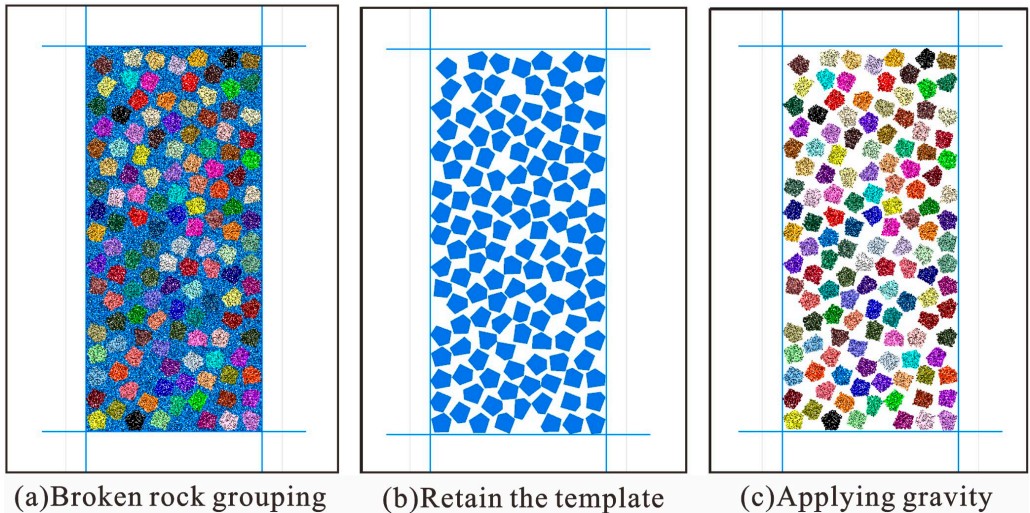

(a)Broken rock grouping　　　(b)Retain the template　　　(c)Applying gravity

**Figure 6.** Schematic of the formation of caved rock mass.

After the generation of the caving rock mass model, gravity settlement was carried out to ensure the full contact of the caving rock masses and to obtain the caving rock mass model under natural accumulation. The compaction process of caving rock masses of different shapes was simulated by controlling the settlement of the pressure-retaining plate. During compaction, the displacement nephogram, the stress–strain curve, the porosity and the fracture rate of broken rock masses of different shapes were monitored. Finally, by analyzing the relationship between the porosity and the compaction characteristics, the water permeability mechanism of the broken rock masses of different shapes during compaction was obtained.

## 3. Test Results of Numerical Simulation

### 3.1. Compaction and Deformation Characteristics of Caving Rock Masses with Different Shapes

The crushed rock was gradually compacted and deformed under the pressure of the overburden rock. Five different prism-number particle clusters—tetragonal, pentagonal, hexagon, octagonal and decagonal—were obtained to establish test samples, and the numerical simulation of lateral compression was carried out. The numerical simulation results are shown in Figure 7.

Through processing the numerical simulation results, the stress–strain curve changes of different shapes of caving rock masses were sorted out, as shown in Figure 8. As shown in Figure 8, the changes in the compaction stress–strain curves of the caving rock masses of different shapes are exponential, and there are both compaction stages (straight section) and nonlinear change stages. This is similar to the compaction stress–strain relationship of broken rock masses under different size and grading conditions [7,8]. When the stress level was low, it was mainly manifested as the compaction stage; the longest stage was for the quadrangular caving rock masses, and the shortest stage was for the decagonal caving rock masses. At this stage, the caving rock masses were loose porous media, and the strain greatly increased when the stress increment was small. With the increase in stress, when the axial strain of the tetragonal caving rocks and ten deformed caving rocks reached approximately 17% and 6%, respectively, a nonlinear stage was gradually shown, in which the stress increment was large and the strain increment was small. Under the same stress conditions, the strain produced by the quadrangular caving rock masses was the largest,

and the ultimate limit strain was larger, while the limit strain of the caving rock masses with more prisms was smaller.

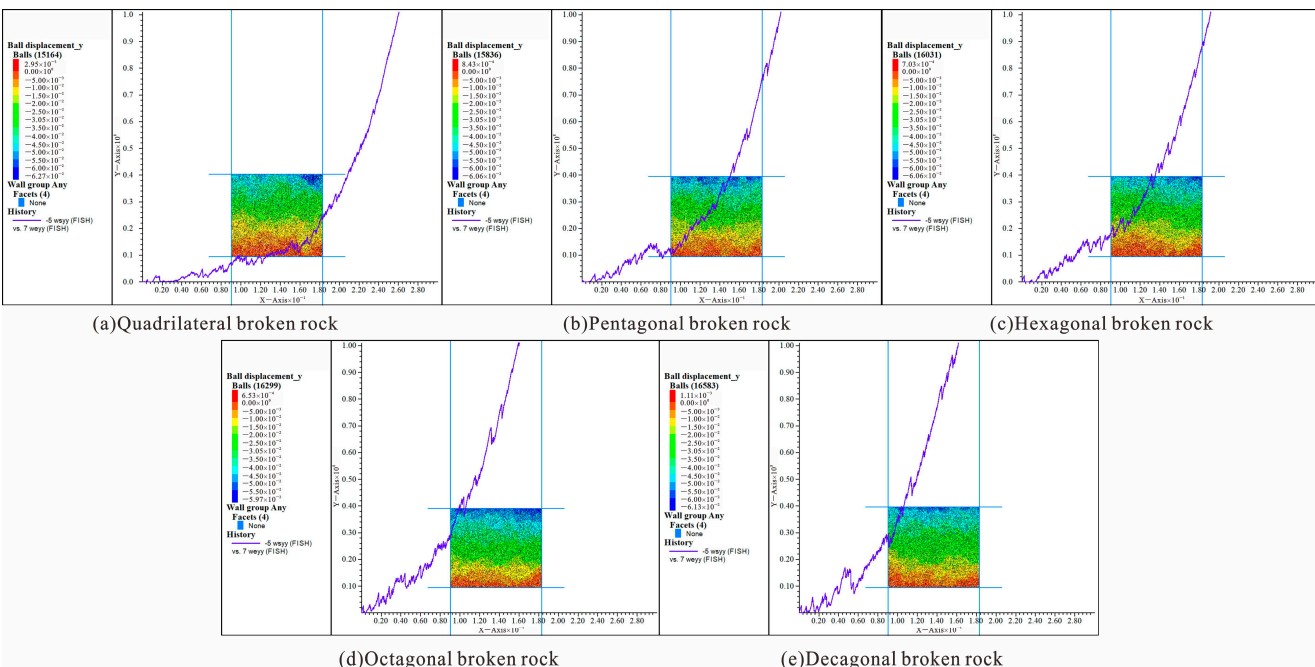

(a)Quadrilateral broken rock     (b)Pentagonal broken rock     (c)Hexagonal broken rock

(d)Octagonal broken rock     (e)Decagonal broken rock

**Figure 7.** Compaction curve and displacement nephogram of broken rock masses with different shapes.

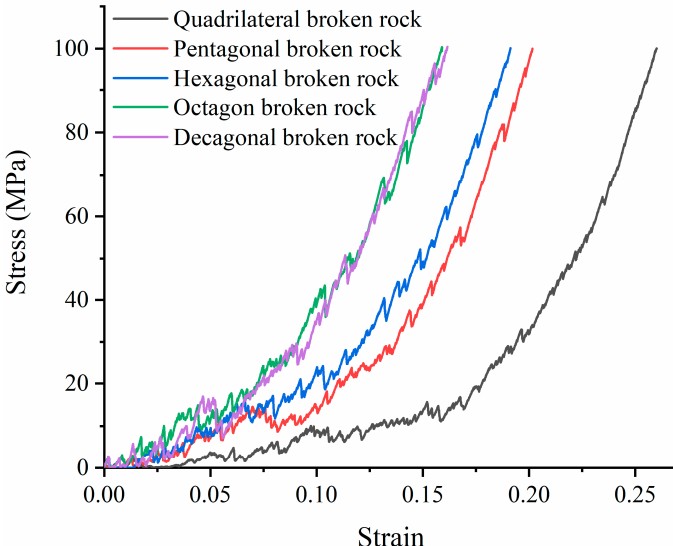

**Figure 8.** Compaction stress and strain curve of caving rock masses with different shapes.

### 3.2. Stress–Strain Relationship and Deformation Modulus of Caving Rock Mass with Different Shapes

The tangent modulus and secant modulus of a broken rock mass are important indexes to measure the deformation resistance of a broken rock mass under compression. Figure 9a shows the relationship of the tangential modulus of the caving rock masses with the strain in different rock mass shapes. The tangential modulus of rock masses of different shapes is exponentially related to the strain [18]. When the strain is small (approximately 8%), the tangential moduli of rock masses with different shapes are not significantly different. Rock masses with few edges are slightly smaller than that with more edges. With a continuous increase in strain, the tangential modulus of the octagonal and decagonal

caving rock masses enters the nonlinear stage and starts to rapidly increase. Later, the caving rock masses with fewer edges enters the nonlinear stage, and when the number of edges continuously increases to a certain extent, the shape resembles a circle, and the shape has minimal influence on the tangential modulus.

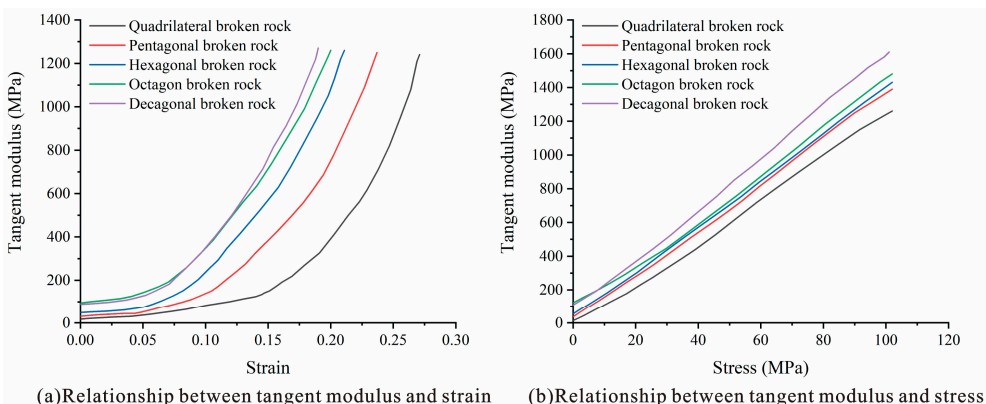

(a)Relationship between tangent modulus and strain       (b)Relationship between tangent modulus and stress

**Figure 9.** Relation curves of tangent modulus, stress and strain of caving rock mass in different shapes.

Figure 9b shows the relationship of the tangential modulus of the caving rock masses with stress in different shapes of rock mass. The tangential modulus of the caving rock masses of different shapes exhibits a slight difference when the stress is small. With a continuous increase in stress, the tangential modulus gradually increases [18]. The tangential modulus of the caving rock masses with a larger number of edges increases at a faster rate than that of the caving rock masses with fewer edges.

Figure 10 shows the relationship of the secant modulus of the caving rock masses with the stress–strain curve. The sectional modulus of rock masses of different shapes is exponentially related to strain [18]. At the beginning of strain, the difference in the sectional modulus of the rock masses of different shapes was obvious, and the initial sectional modulus of rock masses with few edges was small. With a continuous increase in strain, the more edges the caving rock masses had, the earlier their secant modulus entered the acceleration stage. The secant modulus of rock masses with different shapes is linearly related to the stress, and the growth rate of the secant modulus is similar to that of the stress. Under the same stress conditions, the higher the number of edges, the larger the sectional modulus of the caving rock masses. With an increase in the number of edges, the trend gradually weakens.

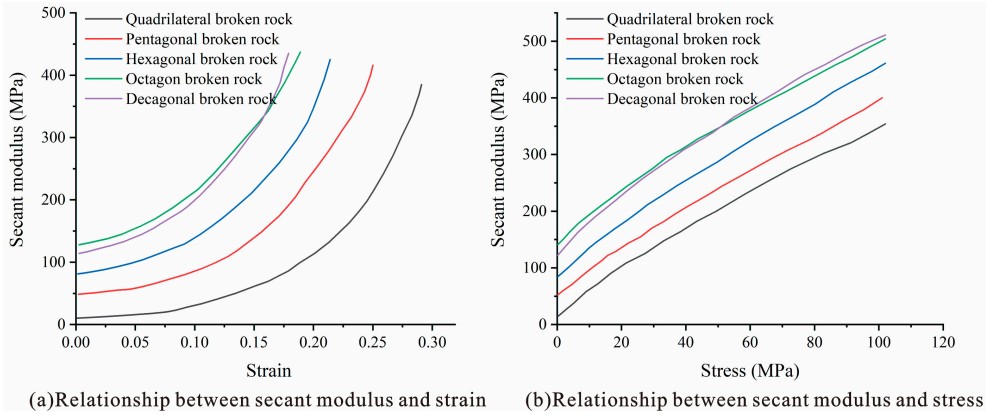

(a)Relationship between secant modulus and strain       (b)Relationship between secant modulus and stress

**Figure 10.** Relation curve of secant modulus, stress and strain of caving rock mass in different shapes.

*3.3. Influence of Different Caving Rock Mass Shapes on the Porosity and the Coefficient of Fragmentation*

(1)　Fragmentation expansion coefficient

The fragmentation of the caving rock masses refers to the increase in the volume of the whole rock after it has been broken, which is usually expressed by the coefficient of fragmentation, $i_0$ [5].

$$i_0 = \frac{v_1}{v_0} \tag{5}$$

where $v_1$ is the volume after the collapse of the caving rock masses and $v_0$ is the original volume [5].

$$v_0 = \frac{m}{\rho} \tag{6}$$

where $m$ refers to the quality of the caving rock masses and $\rho$ is the density of the caving rock masses.

(2)　Porosity

The caving zone of a goaf can be regarded as a porous medium composed of broken rock masses. The continuous subsidence of overlying strata compacting a goaf caving zone results in the change in porosity of the caving zone itself. The change in porosity directly affects the physical and mechanical properties and pore seepage characteristics of the caving zone. Therefore, the characteristics of porosity variation in the compaction process of broken rock masses in the caving zone of a goaf can be accurately identified. It is of great significance to study the movement of overlying strata and the mechanism of surface subsidence.

The porosity of the caving rock masses is the ratio of the pore volume to the total volume (including the pore volume), as shown in Formula (7) [4]:

$$n = \frac{v_1 - v_0}{v_1} \tag{7}$$

where $v_1$ is the volume after the collapse of the caving rock masses and $v_0$ is the original volume.

Figure 11 shows the change curve of porosity with stress and the displacement nephogram of different forms of caving broken rock masses during compaction.

Figure 12 shows the variation curves of the porosity and coefficient of the fracturing expansion of the caving rock masses with stress during compaction. By monitoring the actual volume of the caving rock masses and the dynamic volume change of the test steel, the corresponding porosity curves are obtained. The porosity of caving rock masses of different shapes decreases with increasing stress, and the decreasing trend gradually decreases [26]. When the stress level is approximately 0–20 MPa, the porosity of the quadrangular caving rock masses most obviously decreases, which indicates that the less prismatic the caving rock masses appear, the more obvious is the whole compaction and translation process, and the more obvious is the fracturing phenomenon in the whole process. With a decrease in the gaps in the caving rock masses, the compactness increases, and the decreasing tendency of porosity gradually decreases. The porosity of caving rock masses of different shapes is not significantly different. The porosity curves fluctuate during the process of variation due to the rebreaking of the caving rock masses, which continuously occurs during compaction and recurs until a stable compacted entity is formed.

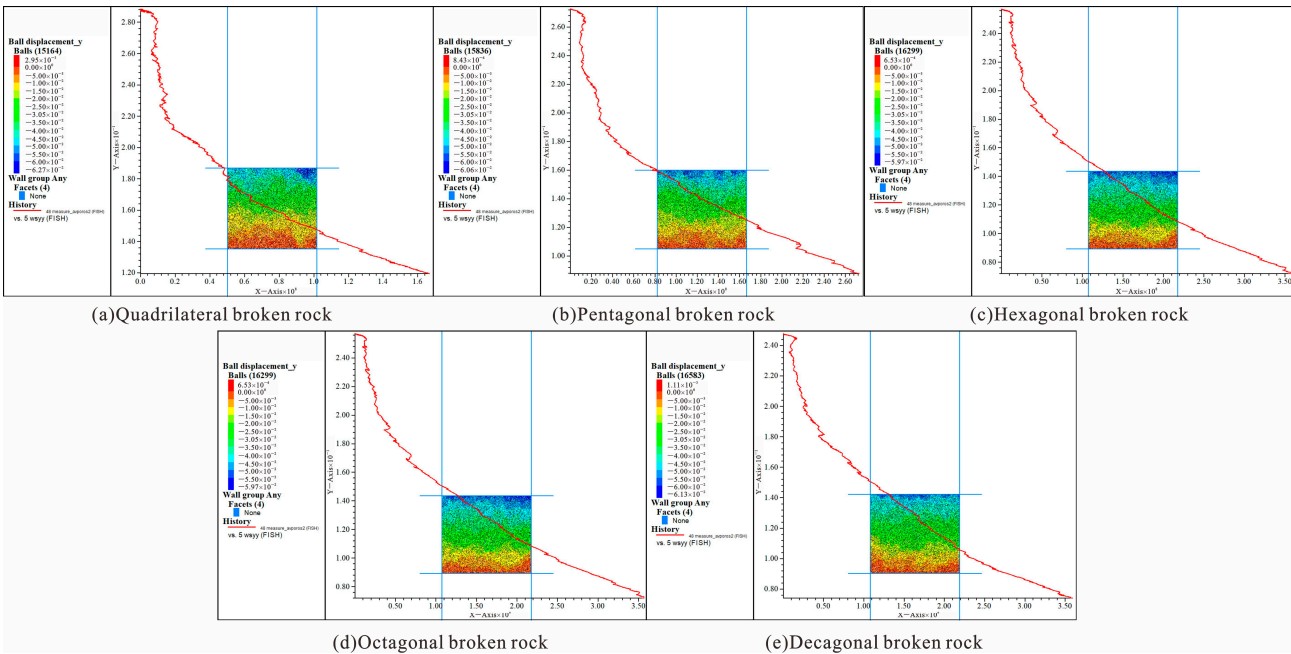

(a)Quadrilateral broken rock  (b)Pentagonal broken rock  (c)Hexagonal broken rock

(d)Octagonal broken rock  (e)Decagonal broken rock

**Figure 11.** Numerical simulation results of porosity and compaction displacement nephogram.

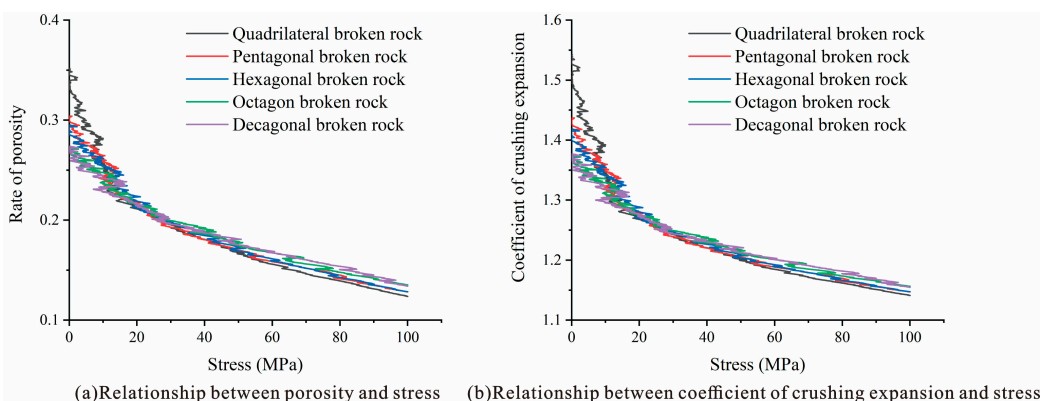

(a)Relationship between porosity and stress  (b)Relationship between coefficient of crushing expansion and stress

**Figure 12.** Relationship curve of porosity, the crushing expansion coefficient and the stress of caving rock masses of different shapes.

Figure 12b shows that the coefficient of the collapse and expansion of the caving rock masses of different shapes varies in the range of 1.15–1.55 as a whole with increasing stress and continuously decreases with increasing stress. The trend of overall change is the same as that of porosity. The coefficient of the initial collapse and expansion of the caving rock masses with few edges is higher than that of rock masses with many edges, but the coefficient of collapse and expansion of those is smaller at the later stage of compaction.

Based on the laboratory test data, the compressive strength of the sandy mudstone was selected as 71.8 MPa [10], and the relationship between the residual fracturing expansion coefficient and the morphology of different caving rock masses was obtained, as shown in Figure 13. The residual fracturing expansion coefficient of the caving rock masses basically remained the same, with a difference of 0.02. With an increase in the number of edges, the coefficient of residual dilation slightly increased because the initial bearing capacity of the rock masses with few edges was lower than that of the rock masses with many edges. When the stress reached the compressive strength, the failure of the rock masses was more serious, while the filling phenomenon was more obvious, resulting in a smaller coefficient of final dilation.

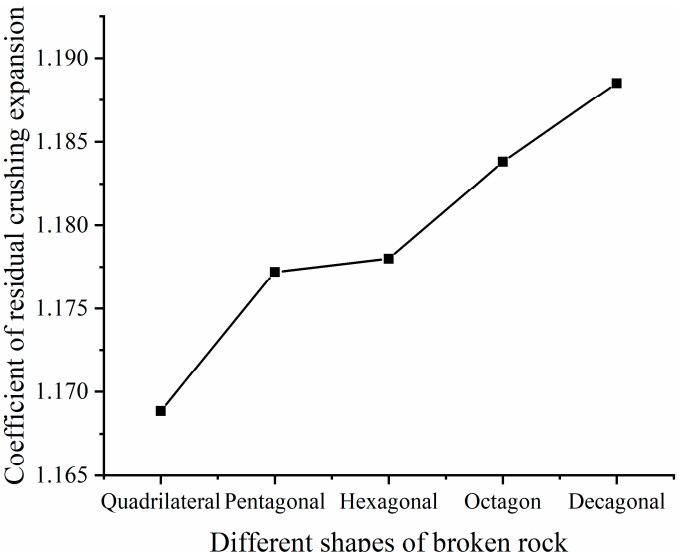

**Figure 13.** Variation curve of residual dilatancy coefficient of caving rock masses of different shapes.

*3.4. Influence of Different Shapes of Caving Rock Mass on the Water Permeability Characteristics*

After coal mining, the broken rock masses in the caving zone of the goaf had the characteristics of irregular size and shape, and sharp edges and corners, and they were easy to break, resulting in differences in the water permeability of different shapes of broken rock masses in the goaf. The difference in the water permeability will affect the seepage of groundwater into the excavation space. On one hand, it easily causes major water inrush and well flooding accidents. On the other hand, it causes the groundwater level of the overlying aquifer to drop, the recharge and the drainage relationship of the groundwater flow system to change, and the ecological environment to be damaged.

The water permeability coefficient can better reflect the hydraulic characteristics and the water permeability of a permeable medium. For porous media, it is generally believed that the water permeability and the porosity evolution can be linked by a cubic mechanism. Therefore, the water permeability can be determined as follows [24]:

$$\frac{k}{k_0} = \left(\frac{n}{n_0}\right)^3 \tag{8}$$

where $k$ is the water permeability coefficient, $k_0$ is the initial water permeability coefficient, $n$ is the porosity and $n_0$ is the initial porosity.

According to the literature, the water permeability ratio is combined with the compaction constitutive mechanism of broken rock mass, and the calculation formula for the water permeability ratio is as follows [24]:

$$\frac{k}{k_0} = \left(\frac{V_0 - \frac{V_S}{1 - \frac{\sigma_Z}{a\sigma_Z + b}}}{V_0 - V_S}\right)^3 \tag{9}$$

where $V_0$ is the initial volume of broken rock masses, $V_S$ is the solid volume, $a$ and $b$ are the fitting parameters and $\sigma_Z$ is the overlying load.

It can be seen from Formula (9) that the size and size distribution of broken rock particles are not directly involved in calculating the water permeability of a goaf, which makes it convenient for us to study the influence of the shape characteristics of broken rock masses on their water permeability. Therefore, the water permeability ratio equation can be used to predict the water permeability of different fractured rock masses.

Figures 14 and 15 shows the relationship curve of porosity with displacement under 10% and 25% strain, respectively. It can be seen from the figure that the porosity of caving rock masses of different shapes decreases slowly when the strain is small. When the strain is large, the decreasing trend of the porosity of caving rock masses of different shapes is approximately linear. Porosity has obvious stage characteristics in different compaction periods. According to the above-described theoretical relationship between porosity and water permeability, it can be seen that the water permeability of broken rock masses of different shapes also has obvious stage characteristics in different compaction periods.

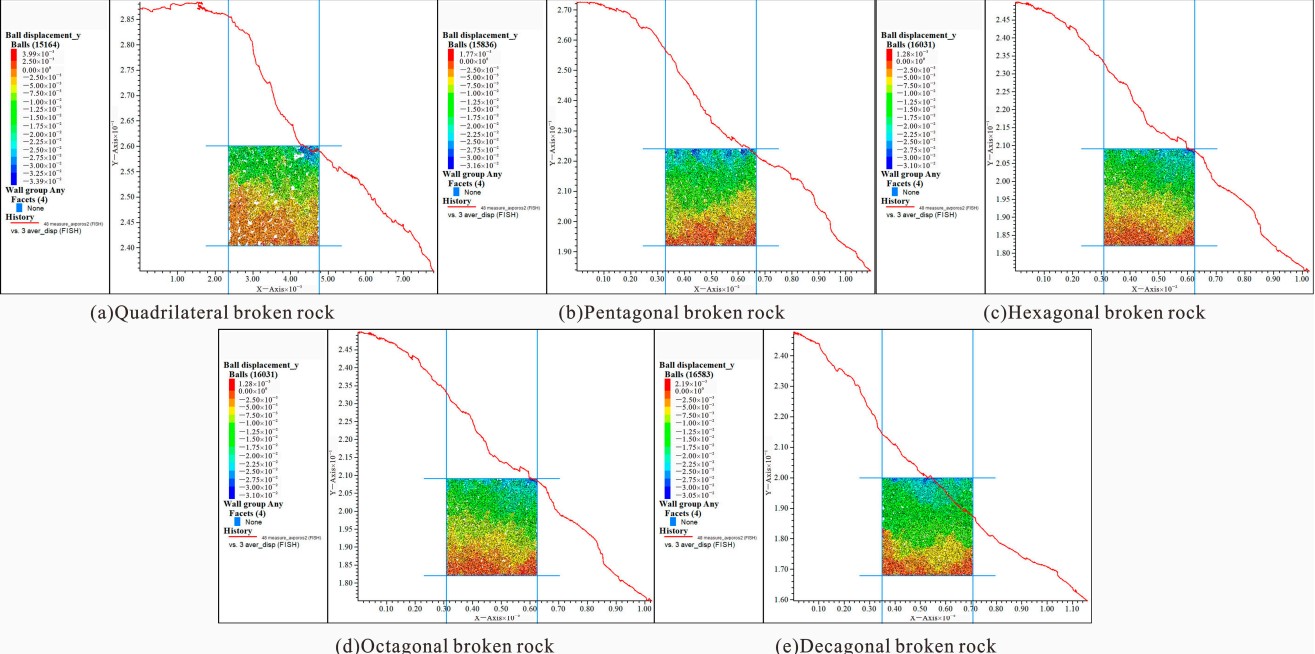

**Figure 14.** Numerical simulation results of porosity at initial compaction stage and compaction displacement nephogram.

As shown in Figure 16, the relationship curve between the water permeability co-efficient ratio and the overlying load is shown. The dotted line in the figure shows the evolution trend of the water permeability coefficient ratio of broken rock masses of different shapes obtained through an empirical formula. The solid line in the figure is the change relationship between the modulus and the pressure brought into numerical simulation, and the evolution trend of the water permeability coefficient ratio of broken rock masses of different shapes is obtained. It can be seen from the figure that the evolution curve of the permeability coefficient ratio obtained by the two methods has good correspondence. The evolution trend of the water permeability coefficient ratio of the fractured rock masses of different shapes is similar, showing a logarithmic relationship. With the increase in stress, the water permeability coefficient decreases gradually. When the stress reaches 80 MPa, the water permeability coefficient ratio changes, becoming stable. The smaller the number of broken rock edges, the faster the ratio of the breaking water permeability coefficient decreases in the initial compaction stage, and the smaller the ratio of the final water permeability coefficient. This is due to the smaller sphericity and irregular shape of the broken rock masses with fewer edges, resulting in less contact points around it and poor strength. When the same normal stress is given to the broken rock masses, stress concentration easily occurs, the broken rock masses are more damaged, and the water permeability coefficient decreases faster. To sum up, the larger the number of edges on broken rock masses, the smaller the loss of the water permeability coefficient under the same compaction state.

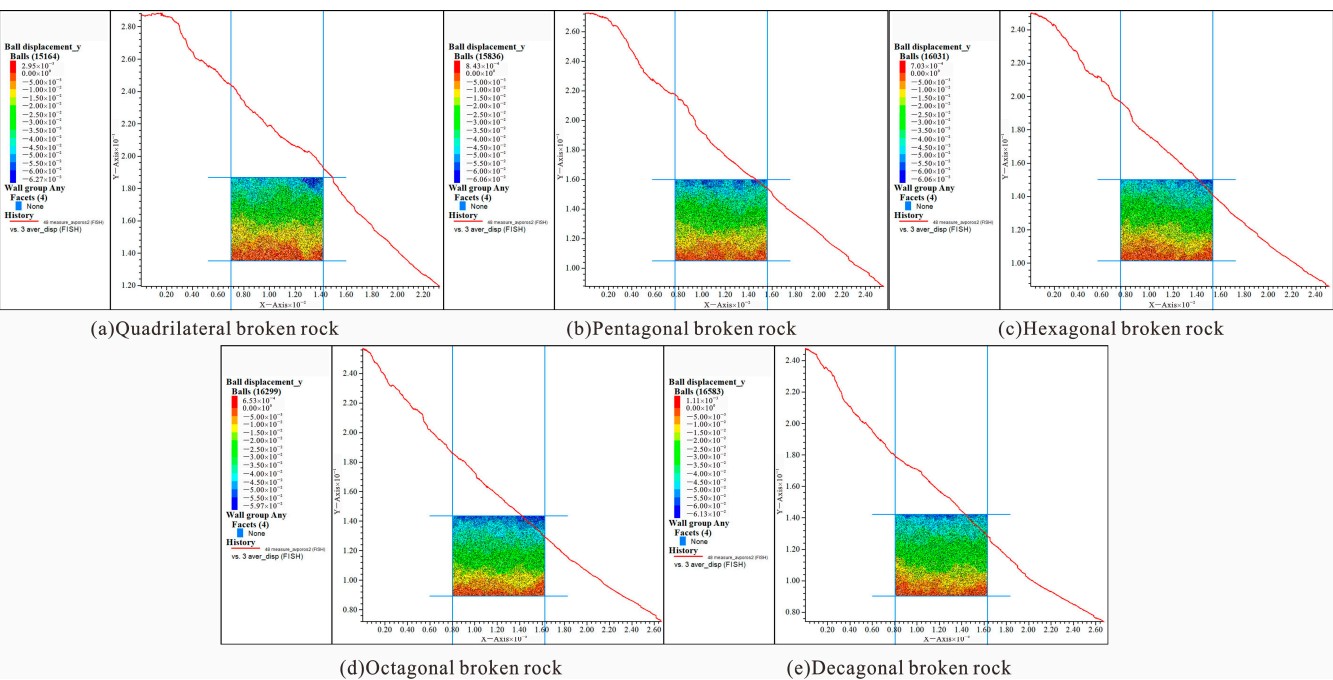

(a)Quadrilateral broken rock  (b)Pentagonal broken rock  (c)Hexagonal broken rock

(d)Octagonal broken rock  (e)Decagonal broken rock

**Figure 15.** Numerical simulation results of porosity and compaction displacement nephogram after compaction.

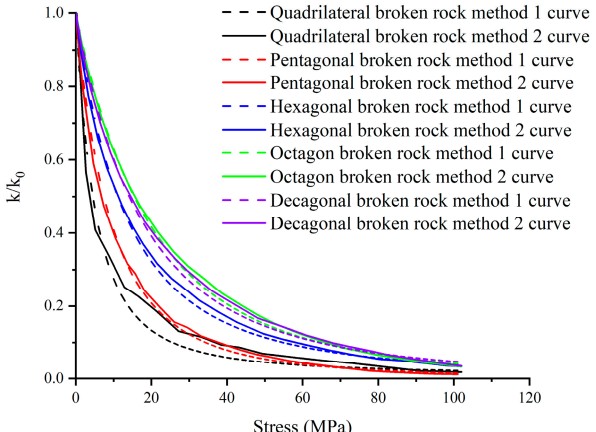

**Figure 16.** Relation curve between water permeability coefficient ratio and overlying load.

### 3.5. Differential Degree of Compacted and Fractured Rocks with Different Shapes

The fracture of the caving rock masses depends on the bond strength and contact strength of the particles. When the stress reaches the critical value, tension or shear failure occurs in the caving rock masses. The maximum tensile stress and shear stress are expressed as:

$$\sigma_{max} = \frac{-F_i^n}{A} + \frac{|M_i^s|}{I}R < \sigma_c \tag{10}$$

$$\tau_{max} = \frac{-F_i^s}{A} + \frac{|M_i^n|}{J}R < \tau_c \tag{11}$$

where $F_i^n$ is the normal stress, $F_i^s$ is the tangential stress, $M_i^s$ is the bending moment, $M_i^n$ is the torque, $A$ is the cross-section area, $L$ and $J$ are the moment of inertia and extreme moment of inertia, respectively, and $R$ is the particle radius between contacts.

As shown in Figure 17, the numerical simulation results of the crushing rate of caving rock masses with different shapes are shown. Figure 18a shows the relationship between

the number of cracks and the strain in different shapes of caving rock masses. The contact between the particles in the caving rock masses is realized by parallel bonding and linear contact. Particle breakage will degrade the parallelly bonded fracture into linear contact. Therefore, the degree of fracture for the caving rock masses is indicated by monitoring the number of parallel bonded fractures. The growth of cracks in different shapes of caving rock masses presents three stages: slow-fast-slow. The more edges there are in the caving rock masses, the earlier the crack number enters the rapid growth stage and the faster the growth rate. When the rock masses are compacted to 80% of their ultimate strain, the growth in the crack number begins to slow again.

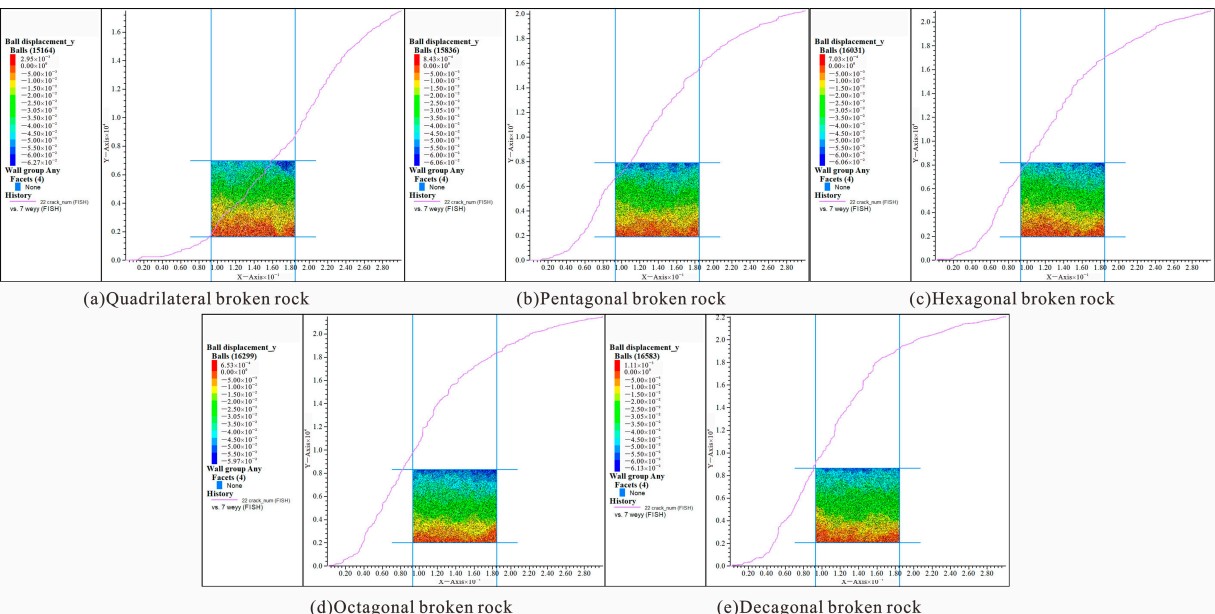

**Figure 17.** Numerical simulation results of crushing rate and compaction displacement nephogram.

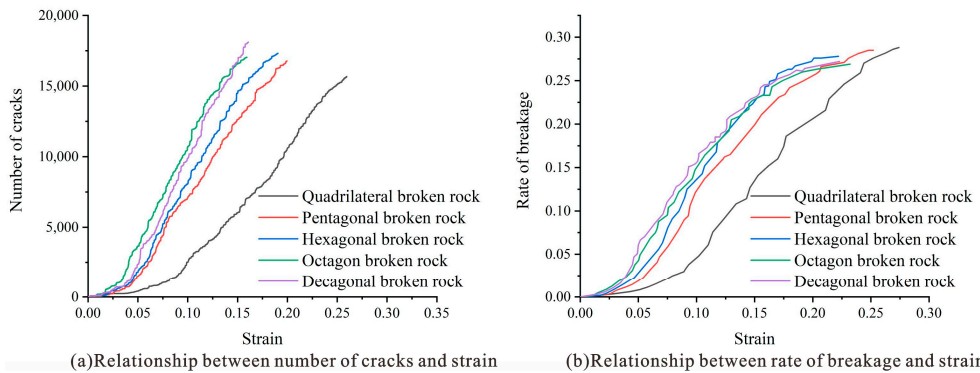

(a)Relationship between number of cracks and strain

(b)Relationship between rate of breakage and strain

**Figure 18.** Relation curve of crack quantity, fracture rate and strain of caving rock mass with different shapes.

Under the same strain condition, the number of cracks in rock masses with few edges is small, but this does not mean that the breakage rate is low. The total contact number of different rock masses with few edges is inconsistent, and the number of particle clusters in rock masses with few edges is small, which results in the least contact of a single rock mass. Therefore, the concept of the breakage rate is introduced, which is expressed as the ratio of the current reduction of the number of parallel bond contacts to the initial parallel bond contact [11]:

$$B_r = \frac{\Delta m}{m} \tag{12}$$

where *m* is the number of parallel bonds and $\Delta m$ is the reduction of parallel bonds, which can be used to visually characterize the degree of fracture of caving rock masses.

The relationship between the fracture rate and fracture strain is shown in Figure 18b. The fracturing rate of rock masses with different shapes changes slowly rapidly slowly with strain [25], and the final failure rate of the rock masses is in the range of 25–30%. Figure 18 shows two inflection points on the curve. The first inflection point is mainly due to the large gap between the broken rock masses at the initial stage of compaction, and the broken rock mainly rotates and moves; at this time, the increase in the breaking rate is small. With increasing stress, broken rock masses are broken and destroyed, leading to a sharp rise in the breaking rate, and the first inflection point of the breaking rate change curve. When it comes to the late stage of compaction, the structural pores of broken rock masses are filled, it is difficult to for new displacement and filling to occur, and integrity is enhanced. Therefore, the growth trend of the breakage rate tends to be flat, and the curve of the breakage rate changes at the second inflection point. With an increase in the number of edges on caving rock masses, the fracturing rate of the caving rock masses decreases. The final fracturing rate of tetragonal rock masses reaches 30%, while the final fracturing rate of decagonal rock masses reaches 26%. The main reason is that with an increase in the number of edges on caving rock masses, the roundness continuously increases. The more contact there is between caving rock masses, the less stress concentration occurs, and the lower is the fracturing rate of the caving rock masses. The ultimate failure rate of rock masses with different shapes does not exceed 30% because the porosity of the rock mass structure is small at this time, it is difficult to displace and fill, and the integrity is enhanced. Thus, the growth trend of the failure rate tends to be flat.

Combining the change trend of porosity and fracturing rate, it can be concluded by analysis that broken rock masses with fewer edges have irregular shapes, smaller sphericity and sharper edges. When broken rocks sustain each other, large pores are formed, resulting in less contact points around them, and in their own strength being poor. When the same normal stress was applied to the rock masses, stress concentration easily occurred, and the stress spread by the single-point contact was greater. Thus, the stress at this point was more easily damaged than its own strength. Therefore, the rock masses with few edges had a higher fracturing rate than those with many edges.

*3.6. Characteristics of Energy Evolution during Compaction and Fracture of Caving Rock Masses with Different Shapes*

Influenced by the overlying load, many defects, such as cracks, often occur in the compaction process of caved and crushed rock masses. The existence of defects weakens the physical and mechanical properties of the caved rock masses, and the stress concentration on the tip of defects is very likely to cause the overall failure of the caved rock masses. The failure of a caving rock mass is a state destabilization phenomenon under the action of energy conversion, which is essentially an irreversible thermodynamic process in which energy dissipation and release are very important. Therefore, it is necessary to analyze the compaction and fracture mechanism of rock masses with different shapes from the angle of energy conversion.

Assuming that there is no heat exchange effect between the test environment and the outside and disregarding the kinetic energy transformed by the ejection of the caving rock masses, the total input energy *U* of the work performed by the external force is the total strain energy. According to the first mechanism of thermodynamics, the calculation formula of the strain energy in the process of lateral compression of the caving rock masses is obtained as follows [8]:

$$U = U_e + U_d \tag{13}$$

where $U_e$ is the elastic strain energy and $U_d$ is the dissipated strain energy. The total strain energy, *U*, absorbed by the caving rock masses is converted to releasable elastic strain energy, $U_e$, stored in the specimen and dissipated strain energy, $U_d$, utilized for damage propagation.

Since only axial stresses work on the specimens during the whole test process, according to thermodynamic theory, $U$, $U_e$ and $U_d$ are obtained by the following formula [8]:

$$U = \int_0^\varepsilon \sigma d\varepsilon \tag{14}$$

$$U_e = \frac{1}{2}\sigma\varepsilon = \frac{1}{2E_0}\sigma^2 \tag{15}$$

$$U_d = U - U_e \tag{16}$$

where $\sigma$ is the axial stress, $\varepsilon$ is strain, and $E_0$ is the modulus of elasticity.

Figure 19 shows the evolution curves of the stress–strain curve, total strain energy ($U$), elastic strain energy ($U_e$) and dissipated strain energy ($U_d$) during the confined compression compaction of broken rocks of different shapes. According to the characteristics of stress–strain curves and crack growth rules, the energy evolution characteristics during the failure process of rock with different shapes are divided into three stages [8]: initial fluctuation growth stage (stage I), relative stable growth stage (stage II) and rapid growth stage (stage III). In Figure 19, three stages of compaction of broken rocks of different shapes are marked on the stress–strain curves with red dots and identified with capital letters $O\sim C$. Although the stresses and stored energy in different stages of compaction and failure process of broken rocks with different shapes are different, the energy variation mechanisms of the caving rocks are similar and have undergone the above three stages.

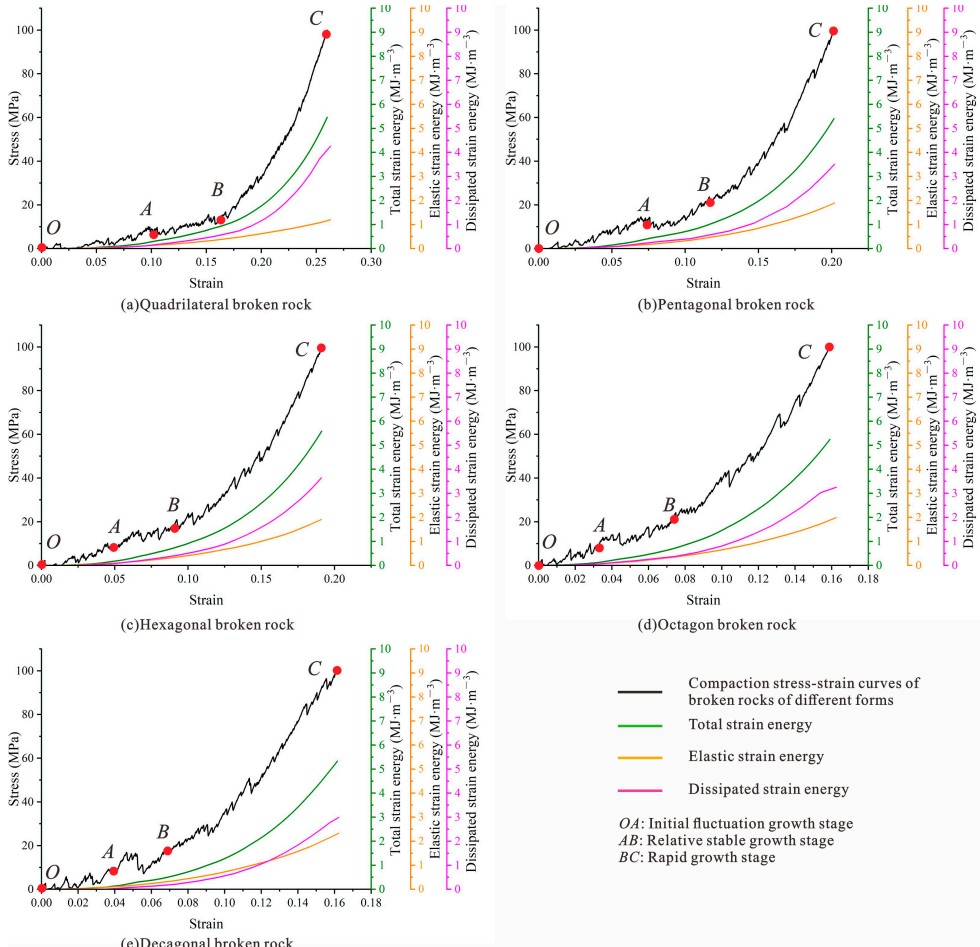

**Figure 19.** Compaction stress–strain curve and strain energy evolution curve of caving rock masses with different shapes.

For the initial wave growth stage (*OA* segment), the broken rock is translated and rotated, the gap between blocks is significantly reduced, and the stress strain curve's *OA* segment is concave. The external force performs work on the broken rock, and the caving rock masses gradually accumulate energy. Due to the small stress, the increase rate of total strain energy, elastic strain energy and dissipative strain energy is small, showing a nonlinear upward trend. At this stage, the strain energy of the caving rock masses is mainly stored in the form of elastic strain energy, and a small part of it is converted to dissipative strain energy, which indicates that, in the initial wave growth stage, the broken rock mainly consumes a part of its capacity by block translation, rotation and other behaviors. However, there are few broken phenomena.

For the relatively stable growth stage (*AB* segment), the *AB* segment of the stress–strain curve changes approximately linearly. With increasing axial stress, the energy stored in the broken rock continuously increases, and its growth rate is greater than that of stage I. Because the collapsed rock masses are mainly broken at this stage, the overall change in the collapsed rock masses is small, and some edges and corners are broken. The caving rock masses continuously undergoes plastic deformation at this stage, so the strain absorbed by the caving rock masses begins to change from elastic strain energy to dissipative strain energy, and the proportion of dissipative strain energy is gradually higher than that of elastic strain energy.

For the rapid growth stage (*BC* section), the stress strain curve of section *BC* shows a nonlinear change. Under the action of higher stress, the specimen still absorbs energy, the total energy curve still shows an upward trend, and its growth rate is far greater than that of stage II. As the caving rock masses mainly break and grind at this stage, the work performed by the external force is mainly consumed in the process of particle rubbing, which is manifested by the continuous expansion and connection of the internal cracks of the caving rock masses, breaking into several small particles or even powders, and an increasing amount of damage, leading to the accelerated growth trend of the dissipated strain energy and of the proportion of the dissipated strain energy.

## 4. Discussion

### 4.1. Change in Particle Movement State in the Caving Rock Mass

Using the movement state of the quadrilateral caving rock masses as an example, Figures 20 and 21 show the movement state of the caving rock masses at the initial stage and the late stage of compaction, respectively. The movement state of the caving rock masses at different positions is different at different stages. Figure 20 shows that in the process of lateral compression, there is a gap between the broken rock mass, and in the process of compaction, the magnitude and direction of the force on each contact point are different, resulting in a compression effect, causing the broken rock mass to rotate. This is consistent with the motion law obtained from the real experiment in the reference Qin et al. [18]. The larger the gap, the larger the space for the broken rock mass to rotate. Many caving rock masses are squeezed by several surrounding caving rock masses, resulting in a state of rotary movement of the caving rock masses. From the location distribution of the caving rock masses, with the downward movement of the loading plate, the caving rock masses in contact with the loading plate initially moves downward as a whole and continuously drives the lower caving rock masses to move, resulting in a slightly higher movement speed of the upper caving rock masses than that of the lower caving rock masses, and the first failure occurs. The rotation mainly occurs in the middle upper part of the caving rock cylinder. The more obvious the rotation movement occurs in places with large pores [18]. The broken rock mass with lower porosity is difficult to continue to move downward. There is basically no rotational movement when friction occurs between broken rock masses.

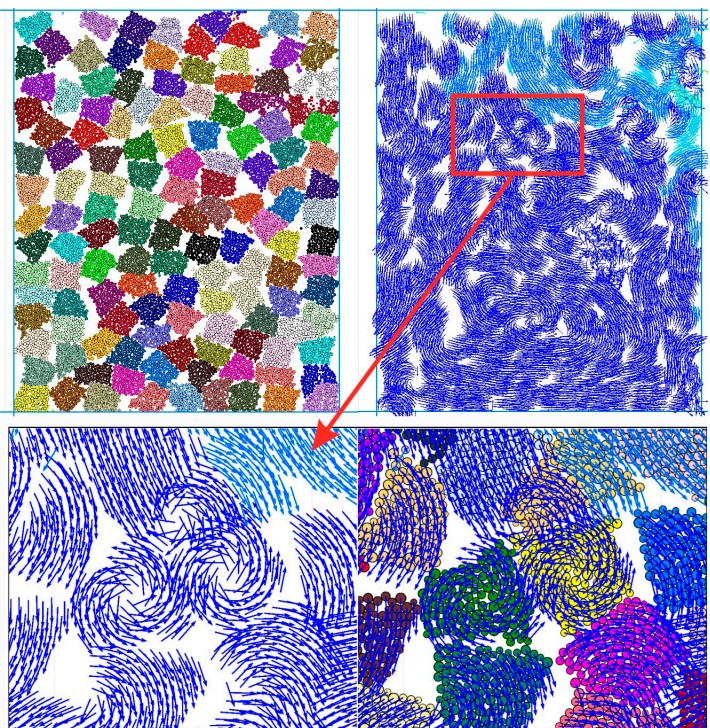

**Figure 20.** Movement state of the caving rock mass at the initial stage of compaction.

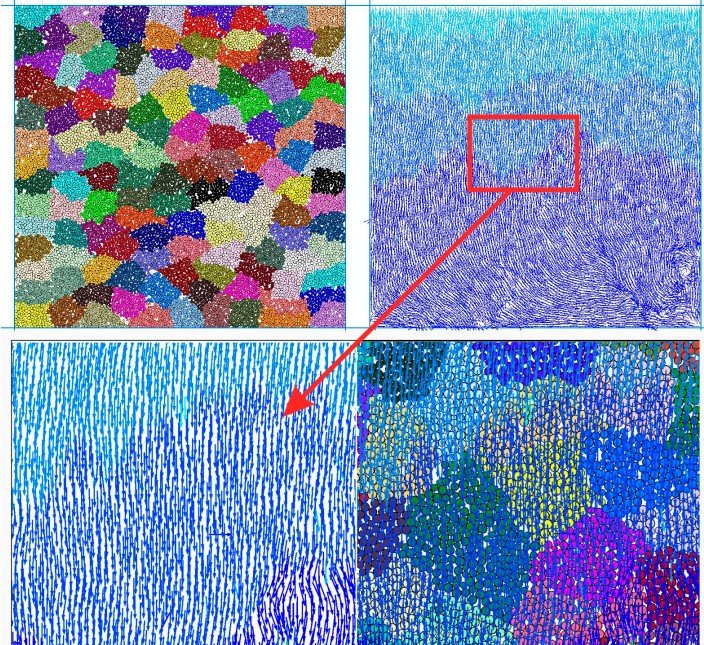

**Figure 21.** Movement state of the caving rock mass in the late compaction period.

Figure 21 shows the movement state of the caving rock masses in the late compaction period. The movement state of caving rock masses tends to be consistent, and the whole range basically presents vertical movement. The surrounding stress of the stable compacted caving rock masses is uniform and the pores are small, so there is no space for rotation, which makes it difficult to rotate again [18]. However, there is still a certain amount of the horizontal velocity component at the lower right side of the caving rock cylinder. The figure shows that there is still a certain amount of porosity at this location, so small

particles broken by the caving rock are forced to move to both sides, and the motion state is relatively complex.

By comparing the movement state of the quadrangular broken rock in different compaction periods, it is concluded that the movement state of the caving rock masses in the test cylinder was not a single vertical compression but gradually changed from the initial rotary filling to vertical compaction. The motion state of other shapes of caving rock masses can be analyzed by introducing the concept of sphericity, $S$, which is defined as follows [27]:

$$S = \frac{r_1}{r_2} \tag{17}$$

where $r_1$ is the maximum inscribed radius of the caving rock masses and $r_2$ is the minimum circumscribed radius.

The morphological characteristics of the caving rock masses with different sphericities are shown in Figure 22. The smaller the number of broken rock edges, the smaller the ratio of $r_1$ to $r_2$. The larger the number of broken rock edges, the smaller the difference between $r_1$ and $r_2$, and the closer the ratio is to 1.0, and the more circular the shape of the broken rock mass. Therefore, caving rock masses with more edges tend to be rounder and more regular in shape. The porosity between broken rock masses is relatively small, the degree of looseness is poor, and there are many contact points. Therefore, caving rock masses have less rotation and translation movement during the compaction process. Caving rock masses have a lower fragmentation rate, and their bearing capacity is better than that of caving rock masses with fewer edges. However, the smaller the number of edges, the more irregular the shape of the caving rock masses, with many gaps between the caving rock masses. During the compaction process, the caving rock masses easily rotate, break, and grind, resulting in an increase in their breaking rate [18].

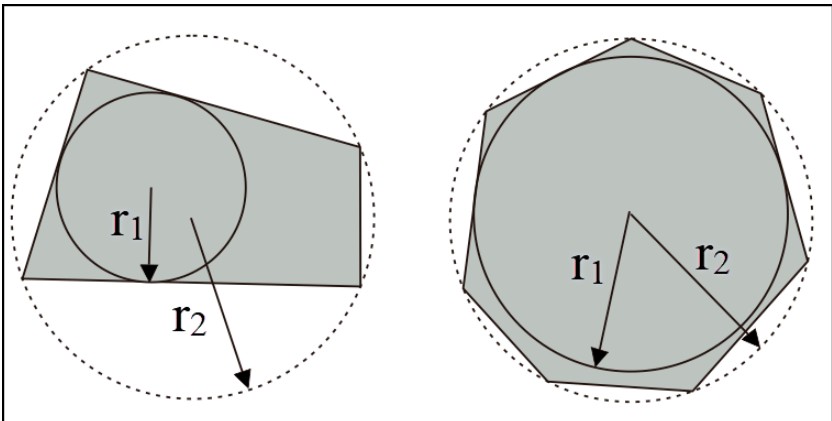

**Figure 22.** Sphericity diagram.

*4.2. Mechanism of the Influence of the Shape of the Caving Rock Mass on its Force Chain Evolution*

Figures 23 and 24 show the dynamic development process of the force chain in the process of confined compression of pentagonal caving rock masses and decagonal caving rock masses, respectively. There is contact force between broken rock masses under external load. The whole contact force distribution network obtained from the statistical summary of the contact forces between all broken rock masses is called the force chain. The thickness of the force chain represents the strength of the force. The force transmission path and direction during the compaction of the caving rock masses can be judged through the force chain distribution diagram to better understand the crushing mechanism and then analyze the crushing mechanism of the caving rock masses.

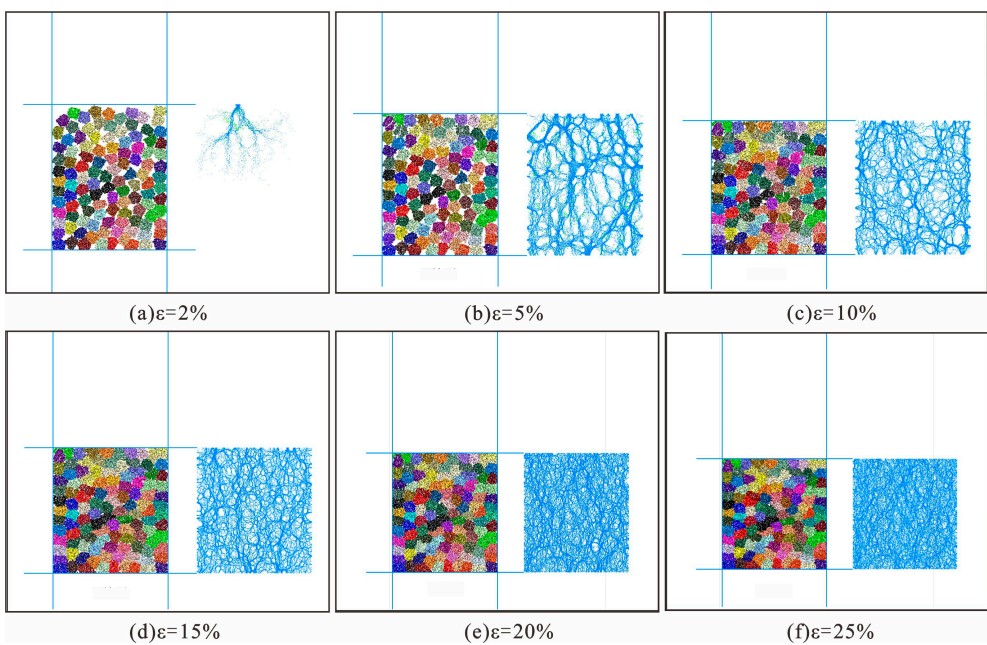

**Figure 23.** Force chain distribution of a pentagonal caving rock mass under different axial strains.

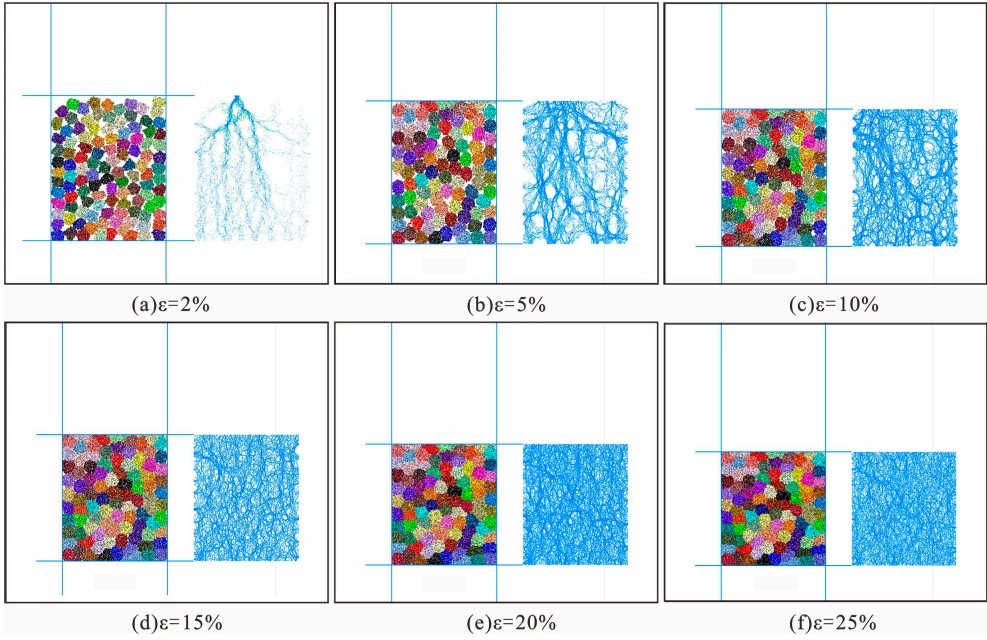

**Figure 24.** Force chain distribution of a decagonal caving rock mass under different axial strains.

Figures 23 and 24 show that with increasing axial strain, the distribution of the force chain in different shapes of rock masses collapses and undergoes obvious changes. Specifically, at the initial stage of compaction, the correspondence between caving rock masses is not high, and the structure is relatively loose. The force chain in the caving rock body is mainly a weak force chain, which is unevenly distributed within the model range. These weak force chains are initially concentrated in caving rock masses that contact the loading plate. With a gradual increase in axial stress and strain, the location of caving rock masses is staggered, the porosity is reduced, and a bearing frame with good contact is gradually formed. The weak-force chain in the upper part of the caving rock masses gradually becomes a thicker and stronger chain, and the force chain gradually extends downward to produce a "skeleton" bearing effect. According to the force chain distribution, the main

direction of the strong chain is consistent with the compaction direction, while the distribution of the weak force chain is relatively random. In the later stage of compaction, most of the caving rock masses have been broken, and the distribution of the caving rock masses is more uniform due to the crushing and filling effect. Most particles in the caving rock masses bear the role of force, and the force chain is evenly distributed. The force chain runs through the whole model, and the shape gradually tends to a stable state. In this state, sufficient contact is established between caving rock masses, making it difficult for particle breakage to continue.

The evolution characteristics of the force chain in the process of compaction are also different for different shapes of caving rock masses. When the strain of pentagonal caving rock masses is approximately 5%, the caving rock masses with large stress start to crack, and initially, the crack mainly appears in the particles at the position of the stronger chain. The stress of decagonal caving rock masses is greater under the same strain condition and mainly occurs during crushing and grinding because the decagonal shape is regular, the roundness is better, the overall bearing capacity of the caving rock masses is greater, and the possibility of damage is small. When the strain reaches approximately 10%, because the pentagonal caving rock masses has many corners, there are many points of contact between the caving rock masses, resulting in the phenomenon of stress concentration, making their force chain distinctly extended. Due to the integrity of decagonal caving rock masses, the force chain distribution in the whole model is more uniform, and more oblique force chains appear. When the strain reaches 15%, the distribution of the force chain in the models of different shapes of rock mass collapse is more uniform. The compaction deformation changes from the movement of the caving rock particles to crushing and filling, and the contact between caving rock masses is closer. Thus, their bearing capacity and deformation resistance are enhanced.

In summary, caving rock masses on the main chain are the key block in compaction, and caving rock masses in the strong chain are destroyed first, and then the particle position is updated to form a new force chain. The largest difference between caving rock masses and clay and fine soil is that the shape and particle size of soil are evenly distributed, the transmission force is more uniform, and there is no significant force chain. The more edges on a broken rock mass, the more regular the shape, and the lower the degree of looseness and porosity between broken rock masses. This leads to more force transmission paths between broken rock masses in close contact. Under high stress levels, irregular caving rock masses are more prone to large deformation if they are in a key position. In practical engineering applications, the proportion of irregular caving rock masses should be reduced.

*4.3. Mechanism of Influence of Caving Rock Mass Shape on the Water Permeability during Compaction*

Figure 25 shows the relationship between the water permeability, the porosity and the stress of broken rock masses of different shapes. It can be seen from Figure 25a that with the gradual increase in stress, the water permeability and the porosity of broken rock masses of different shapes gradually decrease, showing a nonlinear change [14]. From Figure 25b, it can be seen that the water permeability of broken rock masses of different shapes is exponential with the porosity. With the increase in porosity, the water permeability gradually increases, and the growth rate also increases. Under the same porosity, the less edges there are on broken rock masses, the weaker their water permeability is. This is because with the increase in the overburden load, broken rock masses of different shapes move, rotate and break. The pores of broken rock masses are gradually filled, resulting in a reduction in the water permeability and the porosity. Under the same load and the porosity, the smaller the sphericity of broken rock masses, the more irregular the shape. In the process of compaction, the crushing rate is high, resulting in a reduction in the water permeability. Therefore, the impact of broken rock mass shape characteristics on water permeability during compaction should be considered in engineering construction, so as

to reduce water inrush and well flooding accidents, and to timely adjust the recharge and drainage relationship in an underground water flow system.

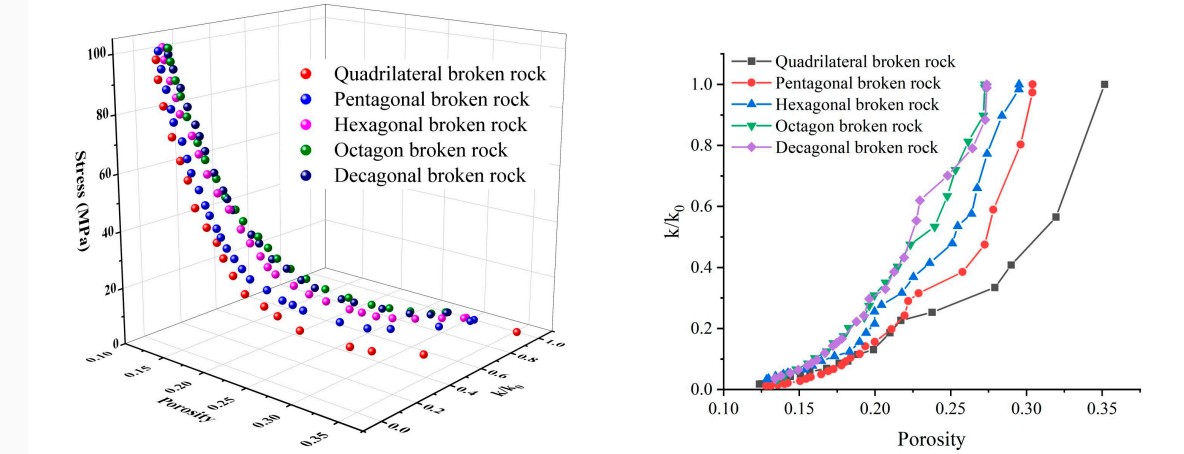

(a)Water permeability – porosity – stress relationship    (b)Relation curve between the water permeability and porosity

**Figure 25.** Relationship between the water permeability, the porosity and the stress of broken rock masses of different shapes.

### 4.4. Mechanism of Influence of Caving Rock Mass Shape on Energy Dissipation during Compaction

Energy dissipation leads to the deterioration of the interior of caving rock masses. The energy release makes the structure of caving rock masses mutate. When the strain energy is rapidly released, caving rock masses are destroyed. Figure 26 shows the changes in strain energy corresponding to different shapes of caving rock masses under the limit strain. As shown in the figure, with an increase in the number of edges on the caving rock masses, the total strain energy and the dissipated strain energy of the caving rock masses show a decreasing trend, and the elastic strain energy shows a growing trend. Under the same stress, the greater the number of edges on the caving rock masses, the stronger their bearing capacity, and the smaller the strain limit, so the lower the total strain energy absorbed. The smaller the number of edges of the caving rock masses, the smaller the sphericity, and the more irregular the shape of the caving rock masses, reducing the amount of contact and contact points between the caving rock masses, resulting in a significant phenomenon of stress concentration, increasing the rate of fragmentation of caving rock masses, and resulting in a sharp release of strain energy, which is converted to dissipative strain energy. However, the collapse of rock masses with a larger number of edges has a lower damage degree, so the elastic strain energy is substantial.

Zhou et al. [8] studied energy dissipation during gangue compaction. Under the same stress condition, the smaller the strain of gangue, the stronger the anti-deformation ability of gangue, and the smaller the energy required for deformation. This is consistent with our findings. Fracture rock masses with more edges have stronger bearing capacity. The less the total strain energy absorbed under the compaction load, the less the total strain energy converted into dissipated strain energy. In conclusion, the evolution of dissipative strain energy degrades the strength of different shapes of caving rock masses and ultimately leads to the failure of caving rock masses of different degrees. Caving rock masses with a larger number of edges have strong integrity, and the higher the compressive strength is, the less the dissipated strain energy is converted during failure, and the lower the degree of damage. Therefore, energy dissipation is directly related to the damage and strength degradation of different shapes of collapsing rock masses during compaction.

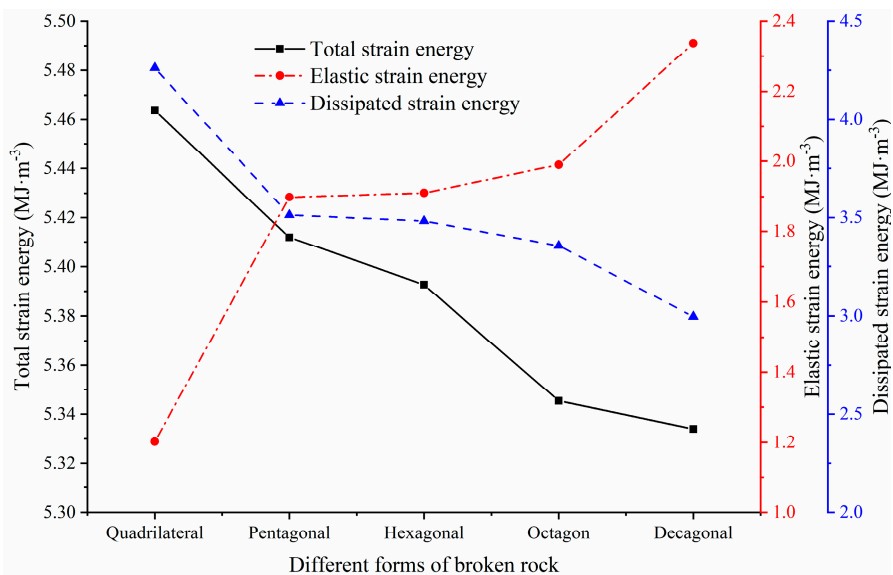

**Figure 26.** Relationship curve of caving rock mass shapes with total strain energy, elastic strain energy and dissipative strain energy.

## 5. Conclusions

This article analyzes the influence of the shape characteristics of the broken rock mass in a goaf on the compaction characteristics and the water permeability of the broken rock mass by simulating a lateral compression test on broken rock masses with different numbers of edges. The main conclusions are presented as follows:

1. The number of edges on a caving broken rock mass is negatively correlated with the limit strain of compaction, the initial void ratio and the final breaking ratio. It is positively correlated with the deformation modulus and the residual dilatancy coefficient.
2. When the shape of caving rock masses is quadrilateral, pentagonal or hexagonal, the initial compaction state of the caving rock masses occurs in a rotary motion, and the later motion state is vertical compression. When the shape of caving rock masses is octagonal or decagonal, the compaction motion of the caving rock masses is mainly vertical compression.
3. The water permeability ratio of broken rock masses of different shapes decreases rapidly at the initial stage, and then gradually reaches a stable stage. The less edges there are on broken rock masses, the faster the rate of decline is in the water permeability ratio, and the lower the final water permeability is.
4. With an increasing number of edges, the total strain energy and the dissipative strain energy of caving rock masses show a decreasing trend, while the elastic strain energy shows a growing trend.

In the simulation process of this paper, the mixing of broken rock masses of different shapes in a certain proportion and their real size have not been considered, which needs to be further improved on the basis of the simulation in this paper. In addition, the loading mode also has an impact on particle breakage. The stress condition of the caving zone in a goaf is not a single uniform loading mode, and the mining stress is not a uniform loading mode. The next step is to study the breaking conditions under this stress state and the change characteristics of the water permeability of the broken rock mass.

**Author Contributions:** Conceptualization, Y.G. and Y.Q.; methodology, Y.G.; software, P.C.; validation, Y.G., Y.Q.; formal analysis, Y.G.; investigation, P.C.; resources, Y.Q.; data curation, Y.G.; writing—original draft preparation, Y.G. and Y.Q.; writing—review and editing, Y.Q. and N.X.; visualization, Y.G.; supervision, Y.Q. and N.X.; project administration, Y.Q.; funding acquisition, N.X. All authors have read and agreed to the published version of the manuscript.

**Funding:** This research was funded by National Natural Science Foundation of China (NSFC) under grant no. 42230709 with the title "Mechanism and prevention methods of tunnel disaster induced by long-term deformation of rock strata in mining-induced subsidence zone" and the title "Urban geological environment and engineering", a high-precision discipline construction project.

**Institutional Review Board Statement:** Not applicable.

**Informed Consent Statement:** Not applicable.

**Data Availability Statement:** Not applicable.

**Acknowledgments:** The authors thank the National Natural Science Foundation of China (NSFC) under Grant No. 42230709 and high-precision discipline construction project.

**Conflicts of Interest:** The authors declare no conflict of interest.

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
