# Peer review of "Simulation of the Compaction Behavior and the Water Permeability Evolution of Broken Rock Masses of Different Shapes in a Goaf"

_water, doi:10.3390/w15061190_

Round 1

Reviewer 1 Report (Previous Reviewer 1)

1.      Concise the abstract to 200 words (limitation of the journal).

2.      English proficiency problems are present in the manuscript (read the second and third sentence, also line 44 (master)). These are just a few examples.

3.      What is the diametric strain?

4.      The stress-strain relationship in figure 1 can be just express in a single line. This figure has no contribution in the later part of the article. Why the authors are including it?

5.      Talbol index? If this is a special index, then cite it.

6.      The authors have concluded the research in the last paragraph of the introduction section and have indicated the research problem. However, the structure of the introduction has some technical writing issues. Also, the research background in the abstract is very general. Be specific and include a sentence in the abstract related to research problem.

7.      According to authors, Voronoi polygon model is used. Discuss its application and why other models like mohr-coulomb is not appropriate.

8.      Table 1 properties are the materials proper of intact rock? can we use the standard UCS tests for the crushed materials?

9.      References for Equations.

10.   Which FISH function the author used in their numerical modelling.

11.   How this study will be beneficial in the real field? Field application?

Author Response

We have uploaded the document of the response reviewer to the attachment. Please see the attachment "Responses to Reviewer # 1".

Reviewer 2 Report (Previous Reviewer 2)

The manuscript discusses the effect of different shapes of caved particles in goaf on the compaction and permeability of fractured rock masses through numerical modeling and theoretical background. The manuscript needs further revisions before it can be processed further. 

The research work assumes that the particles are a typical geometric shape and equally sized. This can have a dramatic effect on the results. Therefore, I would like the authors to explain and justify it.

Consider my comments for improvement (see attached).

Author Response

We have uploaded the document of the response reviewer to the attachment. Please see the attachment "Responses to Reviewer # 2".

Round 2

Reviewer 1 Report (Previous Reviewer 1)

.

Author Response

Thank you very much for the valuable comments put forward by the reviewer. These comments are very valuable for improving the article, so as to make the result of the article convincing.

Reviewer 2 Report (Previous Reviewer 2)

The authors have done a fine job with the revision of the manuscript. All of my comments are addressed and justifications are provided. The authors should add some discussion on shape and size of particles and their influence on caving rock mass, its loose density, porosity etc. 

Also proof read the manuscript to remove minor errors.

Author Response

The reviewer's reply has been uploaded to the file "Responses to Reviewer # 2". Please see the attachment.

This manuscript is a resubmission of an earlier submission. The following is a list of the peer review reports and author responses from that submission.

Round 1

Reviewer 1 Report

1.      The journal instruction for authors are below for the abstract section;

Abstract: The abstract should be a total of about 200 words maximum. The abstract should be a single paragraph and should follow the style of structured abstracts, but without headings: 1) Background: Place the question addressed in a broad context and highlight the purpose of the study; 2) Methods: Describe briefly the main methods or treatments applied. Include any relevant preregistration numbers, and species and strains of any animals used. 3) Results: Summarize the article's main findings; and 4) Conclusion: Indicate the main conclusions or interpretations. The abstract should be an objective representation of the article: it must not contain results which are not presented and substantiated in the main text and should not exaggerate the main conclusions.

Revise your article accordingly.

2.      Split the lengthy sentences and keep the reading fluency in the manuscript.

3.      The statement “Presently, the research results of laboratory tests and numerical simulations on the influence of morphology on the compaction characteristics of broken rock masses are insufficient” is very general. Make the research problem statement specific.

4.      Why “discrete element particle flow numerical simulation” approach is used in this study; justify in the article text.

5.      The two authors participated in the validation. How the authors validated their model? Although figure 5 included for the purpose, however, details are needed.

6.      Figure 4 is the scheme adopted by the authors or the standard procedure. If standard procedure, then need reference. If the authors process, then discuss it in details.

7.      Figures description must be sufficient for self-explanation.

8.      How the authors selected the material properties in Table 1 and 2.

9.      The authors have modelled five shapes, however in real scenario, the material can be mixed with different ratios. Show your results in different ratios and conclude a general trend.

10.   How the study can be extrapolated caving mining field?

11.   The authors have discussed their results comprehensively, however no relevant literature has been cited. Are the authors results are in line with literature or not?

12.   Conclusion must be concise and must be based on the results of this study.

13.   Caving rock mass in goaf is real field scenario. Your study must focus on the field application.

Author Response

We have replied to the reviewer's comments in the document "Responses to Reviewer # 1". Please see the attachment.

Reviewer 2 Report

The paper discusses the numerical modeling of caving. The paper needs revisions before it can be processed. Consider my comments for improvement,.

Author Response

We have replied to the reviewer's comments in the document "Responses to Reviewer # 2". Please see the attachment.

Reviewer 3 Report

1. The originality, accuracy, and completeness of the work are satisfactory.

2. The arrangements of references should be consistent throughout the references list.

3. The authors are suggested to revise some typing errors in the manuscript.

Author Response

We have replied to the reviewer's comments in the document "Responses to Reviewer # 3". Please see the attachment.

Round 2

Reviewer 1 Report

.